# Effect of active tuberculosis on the survival of HIV-infected adult patients who initiated antiretroviral therapy at public hospitals of Eastern Ethiopia: A retrospective cohort study

Tadesse Sime[1], Lemessa Oljira [2], Aboma Diriba[3], Gamachis Firdisa[1], Wubishet Gezimu [1]*

1 Department of Nursing, College of Health Sciences, Mettu University, Mettu, Ethiopia, 2 School of Public Health, College of Health and Medical Sciences, Haramaya University, Harar, Ethiopia, 3 Department of Midwifery, College of Health Sciences, Mettu University, Mettu, Ethiopia

* wubishet151@gmail.com

## Abstract

**Data Availability Statement:** All relevant data are within the paper and its Supporting Information files.

### Background

In resource-limited countries such as Ethiopia, tuberculosis is the major cause of morbidity and mortality among people living with the human immunodeficiency virus. In the era of antiretroviral therapy, the effect of tuberculosis on the survival of patients who are living with human immunodeficiency virus has been poorly understood in Ethiopia. Therefore, this study aimed to determine the effect of active tuberculosis on the survival of HIV-infected adult patients who initiated antiretroviral therapy in public hospitals in Eastern Ethiopia.

### Methods

An institution-based retrospective cohort study was conducted among 566 participants from January 1, 2014, to June 30, 2018. The collected data were entered into EpiData version 3.1 before being exported to Stata version 14 for analysis. A Cox proportional hazard model was used to determine the effect of active tuberculosis on the survival of HIV-infected adult patients who initiated antiretroviral therapy, and a p-value less than 0.05 and a 95% confidence level were used to declare statistical significance.

### Result

Of the 566 patients included in the study, 76 died. The mortality rate was 11.04 per 100 person-years in tuberculosis co-infected patients, while it was 2.52 per 100 person-years in non-tuberculosis co-infected patients. The patients with tuberculosis co-infection had a 2.19 times higher hazard of death (AHR: 2.19; 95% CI: 1.17, 4.12) compared to those without tuberculosis. Advanced clinical stage, low CD4+ cell count, and previous episodes of an opportunistic infection other than tuberculosis were found to be independent predictors of mortality.

**Funding:** The author(s) received no specific funding for this work.

**Competing interests:** The authors have declared that no competing interests exist.

**Abbreviations:** AHR, Adjusted Hazard Ratio; ART, Antiretroviral Therapy; BMI, Body Mass Index; CPT, Co-trimoxazole Prophylaxis Treatment; HIV, Human Immunodeficiency Virus; PLHIV, People living with Human Immunodeficiency Virus; PYO, Person year observation, TB: Tuberculosis; WHO, World health organization.

## Conclusion

Co-infection with tuberculosis at antiretroviral therapy initiation increases the hazard of death approximately two-fold. Hence, we recommend key organizations to enhance the region's collaborative interventional and preventative strategies for TB and HIV.

## Introduction

Human health has been significantly impacted by the human immune virus-tuberculosis (HIV-TB) syndemic, which disproportionately affects people in Africa [1]. In 2015, of an estimated 10.4 million cases of tuberculosis (TB) disease globally, 1.2 million (11%) cases were among people living with human immune viruses (PLHIV), and 60% of TB cases among PLHIV were not diagnosed or treated, resulting in 390,000 TB-related deaths among HIV patients [2]. PLHIV are 15–22 times more likely to develop TB than people without.

TB is the most common illness among PLHIV and it is the major cause of HIV-related deaths, particularly resource-limited settings. Sub-Saharan Africa bears the brunt of the dual epidemic, accounted for approximately 84% of all deaths from HIV-associated TB in 2018 [3]. A recent study conducted on the effect of TB on mortality in PLHIV found that TB increases overall mortality in HIV patients [4, 5].

Ethiopia is one of the 30 nations with a higher prevalence of TB-HIV co-infection worldwide, and in 2016, out of the 36,761 HIV-positive patients who were newly enrolled in antiretroviral therapy (ART), 5.9% had TB as well [6, 7]. According to Ethiopia's 2014 TB-HIV surveillance report, 9.1% of HIV-infected clients newly enrolled in HIV care had active TB. In particular, the percentages of active TB patients in Harar Regional State and Dire Dawa City Administration were 14.2% and 11.5%, respectively [8]. According to some empirical evidence, being co-infected with TB increases the risk of death in HIV-infected patients when compared to those who are not co-infected with TB [9–12].

Regarding the predictors of mortality, scientific evidence has identified a few baseline socio-demographic and clinical predictors of mortality in HIV-infected patients [11, 13–15]. TB mortality among HIV-positive patients is one of the key indicators that measure the impact of TB-HIV collaborative activities [16]. Ethiopia has been implementing national collaborative activities between TB and HIV/AIDS control programs like routine HIV testing among presumptive and diagnosed TB cases; TB screening among PLHIV; early initiation of ART; improved infection control; and provision of TB preventive treatment to decrease the burden of TB in PLHIV [17].

Despite the various evidence across the world showing the causal effect of TB on the mortality of TB-HIV co-infected patients before the ART era, there has been insufficient data to draw firm conclusions about the phenomenon in the ART era [18]. Therefore, this study aimed to determine the effect of TB co-infection at ART initiation on the survival of HIV-infected adults initiated on ART in public hospitals in Eastern Ethiopia. The study's hypothesis was that there is no significant difference in survival between HIV-infected people with and without TB at the start of ART.

## Methods and materials

### Study design, period and area

An institution-based retrospective cohort study was conducted from January 1, 2014, to June 30, 2018, at public hospitals in Harar Regional State and Dire Dawa City Administration of

Eastern Ethiopia. The study included all the public hospitals (Hiwot Fana Specialized University Hospital, Jugol Hospital, Dil Chora Referral Hospital, and Sabian General Hospital) in the two regions.

## Study participants and eligibility criteria

**Inclusion criteria.** PLHIV who started ART from January 1, 2014, to June 30, 2018, were included in the study and classified into TB co-infected and not-infected groups. Patients with TB co-infected were included in the exposed group, whereas those not co-infected with TB were included in the non-exposed group.

**Exclusion criteria.** Patients with incomplete registration cards on the date of ART initiation, patients who started ART from other healthcare institutions (transferred in), and mothers who initiated ART for the prevention of mother-to-child transmission (PMTCT) were excluded.

## Sample size calculation

The sample size for the Cox proportional hazard (PH) regression was calculated using Stata software, assuming a 95% confidence interval (CI), 80% power, 0.05 alpha, 0.2 beta, 20% loss to follow-up, 0.1 overall probability of the event, 2.3 adjusted hazard ratio (AHR) taken from the Somali Region study [11], and a 1:1 exposed to unexposed group ratio, which gives a total sample size of n = 566. Of the 566 patients, 283 were included in the exposed group, whereas 283 were in the non-exposed group. Both groups were selected by a simple random sampling technique using a unique identification number and, after enrollment, retrospectively followed for six months until December 31, 2018 [**Fig 1**].

## Sampling techniques

Of the two regions (Harar and the Somali Regional States) and a city administration (Dire Dawa City Administration) found in eastern Ethiopia, we randomly selected the Dire Dawa City administration and Harar Regional State. All the public hospitals (Hiwot Fana Specialized University Hospital, Jugol Hospital, Dil Chora Referral Hospital, and Sabian General Hospital) found in the two regions were included in the study. A random sampling stratified by hospital and exposure status was used to select the study participants. First, the required sample was proportionally allocated to the four hospitals. Then the population in each hospital was stratified based on their exposure status. Finally, every study participant was selected randomly from each stratum [**Fig 1**].

## Data collection tool, procedures, and quality control

The data collection instrument was adapted from the standardized national ART entry and follow-up form of the Ethiopian Federal Ministry of Health (FMOH) and other similar literatures [**S1 Table**].

Data extraction was done by eight nurses who work in ART clinics and, likewise, supervised by two BSc nurses. The profiles of all selected HIV-infected patients who initiated ART in the period between January 1, 2014, and June 30, 2018, were reviewed; lab requests, follow-up forms, ART intake forms, and patient cards were reviewed. If there were no laboratory tests at ART initiation, results obtained within one month of ART initiation were used as the baseline. The patient's date of death was extracted from the ART follow-up registration form and from the patient death summary sheet for hospital death.

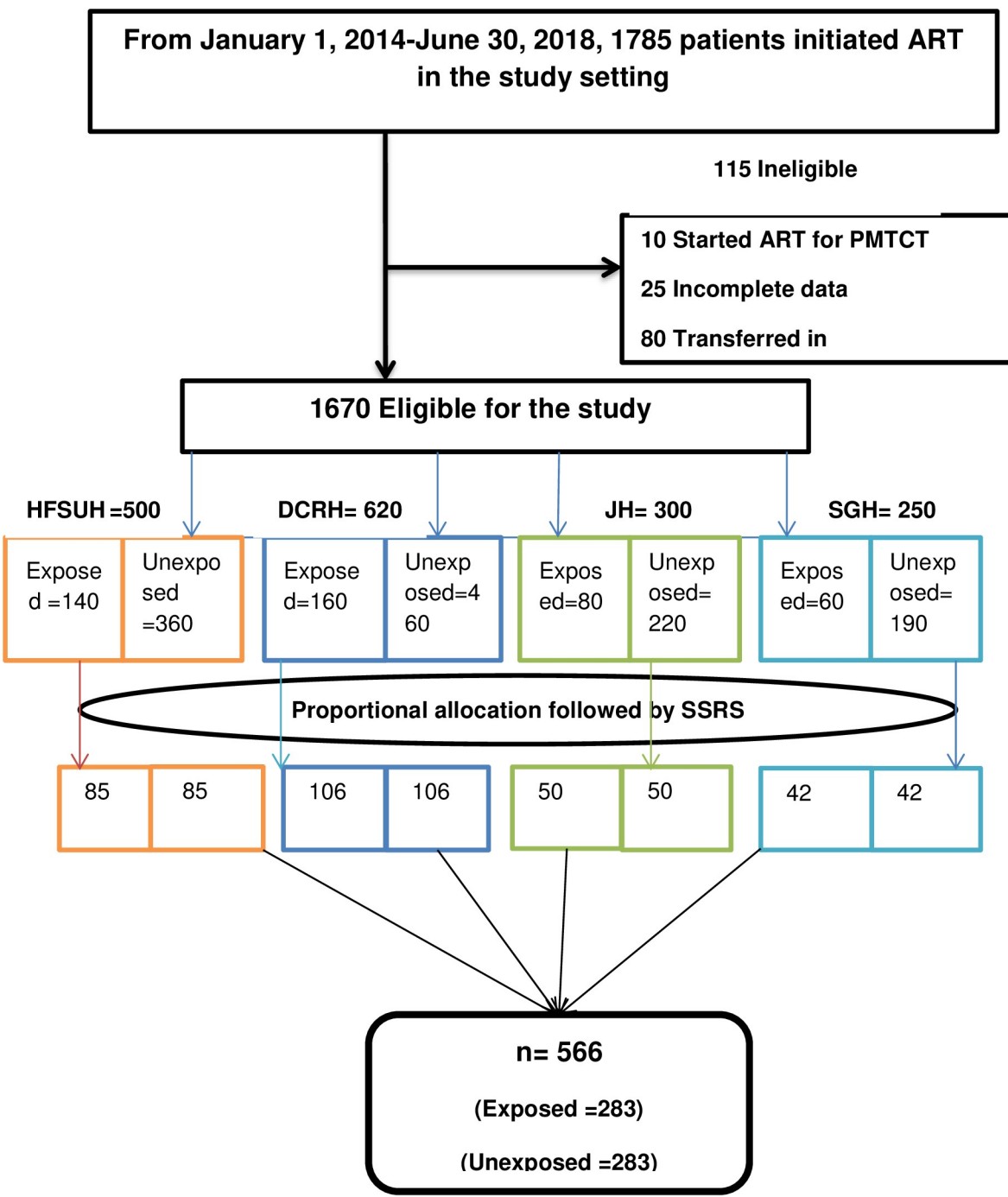

**Note:** HFSUH: Hiwot Fana Specialized University Hospital, JH: Jugol Hospital, SGH: Sabian General Hospital, DCRH: Dil Chora Referral Hospital, SRS: stratified simple random sampling.

**Fig 1. A schematic presentation of patient selection procedures.**

To ensure data quality, training was provided for data extractors and supervisors. Data supervisors, data clerks, and investigators checked the completeness and consistency of data before and after data entry. Moreover, double data entry was performed to prevent data entry errors.

## Study variables

The outcome variable of this study was the time from ART initiation to death due to any cause during the follow-up time. Age, sex, place of residence, marital status, level of education, occupation, World Health Organization (WHO) clinical stage, CD4+ cell count, hemoglobin level, adherence to ART, the status of co-trimoxazole prophylaxis, body mass index (BMI), functional status, and opportunistic infection (other than TB) were independent variables included in this study.

## Operational definitions

**Event:** All-cause mortality ascertained from death certificates if patients died in hospital and from ART registration reported by adherence supporters via phone calls.

**Censor:** Patient was censored on their last visit if they lost to follow up, date of transfer for transferred out or at the end of the study (on December 31, 2018) if they still alive.

**Time scale:** The survival time was calculated in months using the time between the dates of treatment initiation and the date of the event (death) or date of censoring. The maximum and the minimum follow-up were 60 months and 6 months, respectively.

**TB co-infection:** was any confirmed active TB (pulmonary as well as extra pulmonary) at ART initiation or within the first three months of ART initiation. (1) Definitive tuberculosis (TB) is defined as TB confirmed by the GeneXpert test, acid-fast Bacillus (AFB) in sputum or body fluid/tissue, and chest radiography. (2) Presumptive TB: TB diagnosed clinically and anti-TB treatment initiated.

**Adherence to ART:** Adherence to ART was evaluated by the percentage of missed doses documented by the ART physician and ranked as good (if < 5% (< 2 doses of 30 doses or <3 doses of 60 doses), fair (if between 5−15% (3−5 doses of 30 doses or 3−9 doses of 60 doses) or poor (if >15% (>6 doses of 30 doses or >9 doses of 60 doses) as documented by the ART physician.

**Initiation of Co-trimoxazole prophylactic therapy:** Patients who had been taking Co-trim oxazole for more than a month for prophylaxis.

## Patient functional status:

Working: able to perform usual work in or out of the house.

Being ambulatory patient: able to perform activities of daily living.

Being bedridden patient: not able to perform activities of daily living.

## Statistical analysis

The collected data were coded and entered into Epi Data version 3.1 before being exported to Stata software version 14.2 for analysis. The data were cleaned and edited prior to analysis using simple frequencies and cross-tabulation. The clinical and demographic characteristics of the HIV cohort with and without TB co-infection were described. A Chi-square test (Fisher's exact) and T-test were used to compare categorical and continuous variables between the two cohorts, respectively. The survival time was calculated from the date of ART initiated to death among HIV patients who initiated ART from January 1, 2014, to June 30, 2018, and

retrospectively followed for an additional six months until December 31, 2018. The patient exited on the date of the last follow-up if the patient was lost to follow-up or transferred out; December 31, 2018, if the patient is still alive; or the date of death if the patient died. The Kaplan-Meier survival estimate was used to estimate the probability of death and the median time to death among HIV patients co-infected with TB and patients not co-infected with TB at the initiation of ART. The log-rank test was used to test the equality of survival functions between patients with TB and those without TB. A life table was used to calculate the cumulative survival probability of the cohorts. The fitness of the Cox PH model was checked using graphic and Schoenfield residual tests. The predictors were considered when the resultant curves were parallel and the Schoenfield residuals test was statistically insignificant. Moreover, multicollinearity was checked by using a variance inflation factors (VIF) test. A test value of less than 10 was considered to indicate the absence of multicollinearity between the study variables. The multivariable model was built using the backward elimination method of variable selection and confounding was checked. The percentage change in the regression coefficients (β) of less than 20% reveals the absence of confounders [19]. A Cox PH model was used to identify factors associated with mortality, and a p-value < 0.05 declares the significance of the variables at a 95% confidence level.

### Ethical approval and informed consent

Ethical clearance was obtained from the Institutional Health Research and Ethical Review Committee (IHRERC) of Haramaya University College of health and medical sciences. Following the approval, an official letter of cooperation was written to public hospitals in Harar and Dire Dawa towns. Informed, voluntary, written and signed consent was obtained from the medical directors of the hospitals. To preserve patient confidentiality, nurses working in the ART clinics extracted the data from patients' medical records at each hospital. Moreover, no personal identifiers were used on the data collection form.

## Results

### Socio-demographic characteristics of the participants

A total of 566 HIV-infected patients (283 TB co-infected and 283 not TB co-infected cohorts) were followed retrospectively for a median of 18.8 months with an interquartile range (IQR) of 7.0–36.3 months in TB co-infected and 24.3 months (IQR = 13.7–40.2) months among not TB co-infected cohorts. Both cohorts were statistically different for place of residence (p = 0.003), sex (p = 0.003), and occupation (p = 0.001) attributes of the socio-demographic. The majority of study subjects were urban residents, 238(84.1%) and 261(92.2%) in TB co-infected and not TB co-infected cohorts, respectively. The median age of study subjects was 35 years with an IQR of 28–42 years and 34 with an IQR of 28–40 years in TB co-infected cohorts and non-TB co-infected patients, respectively [**Table 1**].

### Baseline clinical and immunological characteristics

In this study, there was a statistical difference in the WHO clinical stage (p<0.001) between the two cohorts. The majority of TB co-infected patients group were in the WHO clinical stage III at ART initiation (202, 71.4%), and among the cohort without TB, only 69(24.4%) were in WHO clinical stage III. Both cohorts were also statistically different in functional status (p<0.001) and BMI level (p<0.001). Patients' CD4+ cell count (p<0.001) and hemoglobin level (p<0.001) also showed a statistical differences between the two cohorts. The majority of patients had at least one episode of opportunistic infection other than TB in the past. Of them,

**Table 1. Socio-demographic characteristics of the study participants who initiated ART in the public hospitals of Eastern Ethiopia (from January 1, 2014, to June 30, 2018), n = 566.**

| Variables | Categories | TB co-infected | Not TB co-infected | X² Value (df) | p- value |
|---|---|---|---|---|---|
| Residence | Urban | 238(84.1%) | 261(92.2%) | 8.955 | **0.003** |
| | Rural | 45(15.9%) | 22(7.8%) | (1) | |
| Age | Mean (±SD) | 35 ± 10 | 35.13 ± 10.92 | - 0.116 | 0. 907 |
| | Median (IQR) | 35(28–42) | 34(28–40) | | |
| Sex | Male | 144 (50.9%) | 107 (37.3%) | 8.784 | **0.003** |
| | Female | 139 (49.1%) | 176 (62.2%) | | |
| Religion | Muslim | 113 (40%) | 87(30.7%) | 5.897 | 0.117 |
| | Orthodox | 147 (52%) | 164(58.7%) | (3) | |
| | Protestant | 20 (7%) | 24(8.5%) | | |
| | Other [4,5] | 3 (1%) | 6(2.1%) | | |
| Marital status | Married | 115(40.6%) | 128(45.2%) | 5.664 | 0.226 |
| | Never married | 71(25.1%) | 50(17.7%) | (4) | |
| | Divorced | 55(19.4%) | 54(19.1%) | | |
| | Widowed | 33(11.7%) | 37(13%) | | |
| | Separated | 9 (3.2%) | 14(5%) | | |
| Level of education | No formal education | 68(24.4%) | 55(19.8%) | 3.765 | 0.288 |
| | Primary | 112(40.1%) | 103(37%) | (3) | |
| | Secondary | 69(24.7%) | 84(30.2%) | | |
| | Tertiary and above | 30(10.8%) | 36(13%) | | |
| Occupation | Merchant | 43(15.2%) | 38(13.5%) | 25.083 | **0.001** |
| | Government employee | 30(10.6%) | 59(20.9%) | (7) | |
| | Non-governmental employee | 16(5.7%) | 6(2.1%) | | |
| | Day laborer | 39(13.8%) | 31(11%) | | |
| | Driver | 8(2.8%) | 12(4.3%) | | |
| | Farmer | 18(6.4%) | 4(1.4%) | | |
| | Jobless | 117(41.3%) | 118(41.8%) | | |
| | Other [b] | 12(4.2%) | 14(5%) | | |

**Note:** others

[a]: Catholic and traditional believers; others

[b]: student, housewife and commercial sex worker; SD: standard deviation; IQR: interquartile range.

156 (55.1%) were from TB co-infected groups, and 126 (42.86%) were from non-TB co-infected groups (p<0.001) [**Table 2**].

## Comparison of survival based on TB co-infection status

There were 76 (13.4%) deaths in this study cohort, with the majority of deaths (50, or 65.7%) occurring in the first 6 months. All the study participants (566) had contributed a total of 13910.8 person-months; 6455.8 person-months in the TB co-infected cohort and 7455 person-months in the non-TB co-infected cohort, respectively. The overall mortality rate was 6.55 per 100 person year observation (PYO) (95% CI: 5.28, 8.16 per 100 PYO). The incidence of death in TB co-infected cohorts was 11.04 per 100 PYO of follow-up (95% CI: 8.64, 13.2 per 100 PYO) and 2.52 per 100 PYO of follow-ups (95% CI: 1.56, 4.2 per 100 PYO) in non-TB co-infected cohorts.

The median time to death during follow-up has shown variation between the two groups. The median time to death was 2.5 months and 12.2 months in the TB co-infected cohort and

**Table 2. The baseline clinical and immunologic characteristics of study participants who initiated ART in the public hospitals of Eastern Ethiopia (from January 1, 2014, to June 30, 2018), n = 566.**

| Variables | Categories | TB co-infected | Not TB co-infected | $X^2$ Value (df) | p- value |
|---|---|---|---|---|---|
| **WHO clinical stage** | Stage I/II | 13(4.6%) | 185(65.4%) | 230.367 | < 0.001 |
| | Stage III\ | 202(71.4%) | 69(24.4%) | (2) | |
| | Stage IV | 68(24%) | 29(10.3%) | | |
| **BMI** | <16kg/m$^2$ | 66 (23.4%) | 29(10.4%) | 41.798 | < 0.001 |
| | 16–18.5kg/m$^2$ | 105 (37.2%) | 66(23.6%) | | |
| | > = 18.5kg/m$^2$ | 111(39.4%) | 185(66%) | | |
| **Functional status** | Working | 140(49.5%) | 221(78.1%) | 53.430 | < 0.001 |
| | Ambulatory | 96(33.9%) | 50(17.7%) | (2) | |
| | Bedridden | 47(16.6%) | 12(4.2%) | | |
| **Hemoglobin** | Mean (± SD) | 11.3 ± 2.46 | 12.5 ± 2.52 | -5.453 | < 0.001 |
| | Median (IQR) | 11.4(9.7–13.2) | 12.6 (11–14.4) | | |
| **CD4+ cell count level in cells/mm3 (n = 552)** | Mean (± SD) | 189.8 ± 178.2 | 352.9 ± 265.6 | -8.475 | < 0.001 |
| | Median (IQR) | 137 (69–262) | 304(137–489) | | |
| **Initial ART regimen** | 1c = AZT+ 3TC + NVP | 4 (1.4%) | 5 (1.8%) | - | 0.395 |
| | 1d = AZT+ 3TC + EFV | 1 (0.4%) | 3 (1%) | | |
| | 1e = TDF + 3TC + EFV | 276 (97.4%) | 271 (95.8%) | | |
| | 1f = TDF + 3TC + NVP | 1(0.4%) | 3 (1%) | | |
| | 1g = ABC_+3TC_+EFV | 1(0.4%) | 1(0.4%) | | |
| **ART adherence** | Good | 229 (82.1%) | 250 (88.7%) | 6.015 | 0.049 |
| | Fair | 13(4.7%) | 12 (4.3%) | (2) | |
| | Poor | 37(13.3%) | 20 (7%) | | |
| **Co-trimoxazole prophylaxis** | Yes | 257(90.8%) | 235(83%) | 7.524 | 0.006 |
| | No | 26(9.2%) | 48(17%) | (1) | |
| **OPI other than TB** | Yes | 156(55.1%) | 203(71.7%) | 16.824 | < 0.001 |
| | No | 127(44.9%) | 80(28.3%) | (1) | |
| **Follow-up outcome** | Dead | 60(21.2%) | 16(5.7%) | 31.468 | < 0.001 |
| | Transferred out | 18(6.4%) | 18(6.4%) | (3) | |
| | Loss to follow up | 25(8.8%) | 21(7.4%) | | |
| | Alive / On treatment | 180(63.6%) | 228(80.6%) | | |

Note: OPI: opportunistic infection, df: degree of freedom.

non-TB co-infected cohort, respectively. Based on actuarial life table analysis, the estimated survival probability of the whole cohort at 6, 12, 24, 36, 48, and 60 months was 91%, 89.4%, 86.2%, 84.6%, 84.6%, and 80%, respectively. The survival probability at 1 year of ART initiation was 81.6% (95% CI: 76.4% to 85.7%) and 97.1% (95% CI: 94.1% to 98.5%) among those with TB and without TB, respectively. The cumulative survival probability at 60 months of follow-up was approximately 69% (95% CI: 57%-78.9%) in TB co-infected groups and 93% (88.5% -95.6%) among non-TB co-infected groups. The overall survival probability in the TB co-infected was significantly lower than in the non-TB co-infected group throughout the study period (log-rank statistic = 31.01, df = 1, P 0.001) [**Fig 2**].

## Predictors of mortality

After adjusting for WHO clinical stage, past opportunistic infection other than TB, and CD4 + cell count, the hazard of death was more than two-folds (AHR: 2.19: 95% CI 1.17, 4.12)

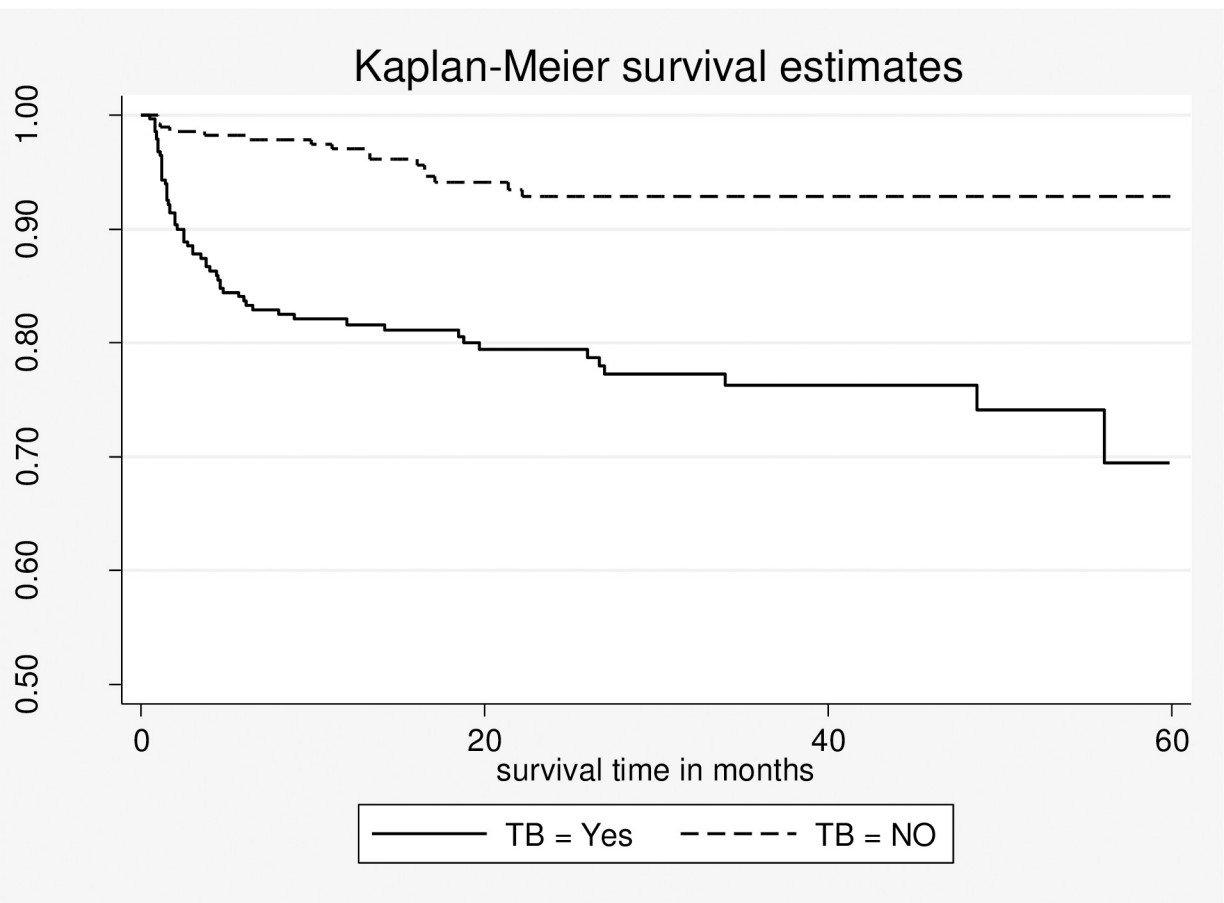

**Fig 2. Kaplan-Meier estimates of survival probability for TB co-infected and not TB co- infected HIV patients who initiated ART in the public hospitals of Eastern Ethiopia (from January 1, 2014 to June 30, 2018) n = 566.**

higher among TB-co-infected patients at ART initiation than those without TB at ART initiation in multivariable Cox regression analysis. Patients in WHO clinical stage IV had nearly three times the hazard of death (AHR 3.06, 95% CI: 1.16, 8.09) than those in WHO clinical stage I/II. Patients with a CD4+ cell count of less than 50 cells/mm3 at the start of ART had a nearly fourfold increased hazard of death (AHR 3.75; 95% CI: 2.00, 7.03) when compared to those with a CD4+ cell count greater than 200 cells/mm$^3$. Moreover, the hazard of death was 65% (AHR 1.65, 95% CI: 1.01, 2.68) higher among patients with at least one opportunistic infection other than TB compared to those without an opportunistic infection other than TB [**Table 3**].

## Discussion

In the current study, the overall mortality rate was 6.55 per 100 PYO in the entire follow-up period. This finding is comparable with previous studies conducted in the seven University Teaching Hospitals and the Somali Regional State of Ethiopia [11, 13]. However, it is higher compared to the studies conducted in Jinka Hospital, South Omo zone of Ethiopia, and other countries of Africa [14, 20, 21]. The reason for this difference might be due to the fact that in this study, over 65% of study participants started antiretroviral therapy while at WHO stage III or IV of the disease severity, and the small number of TB co-infected patients in latter studies.

**Table 3. Results of the multivariable Cox regression analysis of mortality in study participants who initiated ART in the public hospitals of Eastern Ethiopia (from January 1, 2014, to June 30, 2018), n = 552.**

| Variables | Categories | Survival status | | Crude HR (95%CI) | Adjusted HR (95%CI) | P-value |
|---|---|---|---|---|---|---|
| | | Dead | Censored | | | |
| **TB co-infection** | Yes | 60 (79%) | 223 (45.5%) | 4.22(2.43, 7.32) | **2.19 (1.17, 4.12)** | **0.014** |
| | No | 16 (21%) | 267 (54.5%) | (1) | (1) | |
| **Sex** | Male | 40(52.6%) | 209 (42.6%) | 1.45 (0.93, 2.28) | 1.41 (0.87, 2.27) | 0.154 |
| | Female | 36(47.4%) | 281 (57.4%) | (1) | (1) | |
| **WHO clinical stage** | Stage I/II | 7 (9.2%) | 191 (39%) | (1) | (1) | |
| | Stage III | 42 (56.3%) | 229 (46.7%) | 4.79(2.15,10.66) | 1.80 (0.72, 4.52) | 0.206 |
| | Stage IV | 27 (35.6%) | 70 (14.3%) | 9.26(4.03,21.28) | **3.06 (1.16, 8.09)** | **0.024** |
| **BMI** | <18.5 kg/m2 | 49 (66.2%) | 228 (46.3%) | 2.21 (1.37, 3.58) | 1.54 (0.93, 2.55) | 0.091 |
| | > = 18.5kg/m2 | 25 (33.8%) | 260 (53.3%) | (1) | (1) | |
| **Functional status** | Working | 36 (47.3%) | 325 (66.3%) | (1) | (1) | |
| | Ambulatory | 20 (26.3%) | 126 (25.7%) | 1.47(0.85, 2.54) | 0.79 (0.44–1.42) | 0.446 |
| | Bedridden | 20(26.3%) | 39 (8%) | 4.40 (2.55, 7.62) | 1.70 (0.90, 3.23) | 0.101 |
| **CD4+ cell count level in cells/mm3** | < 50 | 23 (30.7%) | 49 (10.2%) | 6.23(3.39,11.46) | **3.75 (2.00, 7.03)** | **<0.001** |
| | 50–199 | 33 (44%) | 160 (33.5%) | 2.86(1.63, 5.04) | 1.78 (0.99, 3.19) | 0.051 |
| | > = 200 | 19 (25.3%) | 268 (56.2%) | (1) | (1) | |
| **ART adherence** | Good | 58 (77.3%) | 421 (86.6%) | (1) | (1) | |
| | Fair | 5 (6.7%) | 20 (4.1%) | 1.73(0.69, 4.32) | 2.08 (0.79, 5.44) | 0.136 |
| | Poor | 12 (16%) | 45 (9.3%) | 2.18(1.17, 4.07) | 1.60 (0.84, 3.03) | 0.146 |
| **OPI other than TB** | Yes | 43 (56.4%) | 164 (33.5%) | 2.45(1.56, 3.86) | **1.65 (1.01, 2.68)** | **0.044** |
| | No | 33 (43.4%) | 326 (66.5%) | (1) | (1) | |

**Note**: 1 = reference category; BMI: body mass index; HR: hazard ratio

In this study, the mortality rate among TB co-infected and those without TB was 11.04 per 100 PYO and 2.52 per 100 PYO, respectively. This finding showed a higher mortality rate among TB co-infected cohorts when compared to a finding from South Africa [22], in which the mortality rate among TB co-infected patients on ART was 4.84 per 100 PYO. This difference might be due to the fact that the country has made robust efforts to tackle the two diseases simultaneously and has stipulated the integration of HIV and TB services nationwide through the co-location of services. Additionally, the variation may be due to the two countries' diverse socioeconomic situations.

In the present study, the estimated survival probability of the cohort at the end of the follow-up period was 80%, and the majority (65.7%) of deaths occurred in the first six months of ART initiation. The survival probability at 1 year of ART initiation was 81.6% among the cohort with TB, whereas it was 97% among those without TB. This finding is in agreement with a retrospective cohort study conducted in the United Kingdom [4]. However, this survival probability is lower than that of a study conducted in the Somali region in which the overall survival probability was 85.9% [11]. This difference might be due to less follow-up time and a higher loss of follow-up in the latter study. On the other hand, the death rate in the first six months was comparable to retrospective cohort studies conducted in Ethiopia [11, 13].

In our study, the hazard of mortality was two-fold higher in patients who were TB co-infected at ART initiation. This finding is consistent with the studies conducted in the United States of America (USA), Uganda, and Ethiopia [5, 11, 12]. This result contrasts with research from South African and Ethiopian studies that found no link between active TB at the start of ART and patients' overall survival [23, 24]. This difference might be due to variations in the

study settings and the level of care given. The country has also provided integrated HIV/TB services for people in correctional facilities, the communities surrounding the industry, and community-based TB-HIV screening and early linkage to care where transmission of both HIV and TB is high. Moreover, the difference might be due to the small number of TB co-infected patients who participated in the latter study.

Regarding the predictors, the present study revealed that patients presenting with an advanced disease stage (WHO clinical stage IV) had a nearly three times higher hazard of death as compared to patients not in an advanced disease stage (WHO clinical stage I and II). This finding is in line with a study conducted in Tanzania [21] and it is also congruent with the previous studies conducted in Ethiopia [11, 14, 25, 26]. This might be due to the fact that HIV weakens the immune system and thereby leads to more opportunistic infections likely to occur, which adversely affect the survival of HIV patients.

In this study, having a lower CD4+ cell count was found to be an independent predictor of death. When comparing patients with a CD4+ cell count of less than 50 cells/mm$^3$ to those patients with a CD4+ cell count of greater than 200 cells/mm$^3$, the hazard of death nearly qua-drupled. This result is consistent with previous studies conducted in Ethiopia [11, 26, 27]. This might be due to the fact that late ART initiation is a pervasive problem in sub-Saharan Africa, which results in poor prognosis and an increased risk of opportunistic infection [28].

In our study, patients who suffered from opportunistic infections other than TB had a 65% higher hazard of death as compared to those patients who were free of opportunistic infections in the past. When compared to a study conducted in the United Kingdom, which discovered that prior opportunistic infections increased the risk of death by 7.4 fold [4], this effect is less pronounced. This difference might be due to the difference in baseline exposure to ART; in this study, patients were followed after ART initiation, while in the later study, follow-up of patients were started from HIV diagnosis.

## Limitation of the study

This research has a few limitations. First, in observational cohort studies, uncontrolled con-founding can never be completely eliminated. Although all deaths were determined to be related to disease and not accidental based on patient death certificates and telephone conver-sations with ART adherence support, the cause-specific mortality was not reported due to a lack of cause-specific data. Thus, the death rate might have been slightly overstated by this con-dition. Factors like patient social conditions were left out of the analysis since they were insuffi-cient for the majority of patients. This result might be slightly biased as a result of the exclusion of patients who transferred out.

## Conclusion

This study showed that HIV patients who were co-infected with TB at ART initiation experi-enced substantial mortality rates. In HIV-infected patients who initiated ART in the study area, having active TB at the time of ART commencement substantially doubles the chance of death. The study also showed that past experiences of opportunistic disease, low CD4+ cell count, and advanced clinical WHO stage (stage IV) were independent predictors of mortality. Therefore, we recommend that key organizations to improve the region's collaborative inter-ventional and preventative programs for TB and HIV.

## Supporting information

**S1 Table. Questionnaire.**
(DOCX)

**S2 Table. STROBE statement checklist.**
(DOCX)

**S1 Data.**
(DTA)

## Acknowledgments

We gratefully acknowledge and appreciate the ART clinic staff and medical record room staff of Hiwot Fana Specialized University Hospital, Jugol Hospital, Dil Chora Referral Hospital, and Sabian General Hospital for their invaluable cooperation during data collection. We are also grateful to the Harar Regional Health Bureau and Dire Dewa City Administration Health Bureau for facilitating conditions while carrying out this study.

## Author Contributions

**Conceptualization:** Tadesse Sime, Lemessa Oljira.

**Data curation:** Tadesse Sime, Lemessa Oljira, Aboma Diriba, Gamachis Firdisa.

**Formal analysis:** Tadesse Sime, Wubishet Gezimu.

**Investigation:** Tadesse Sime, Lemessa Oljira, Aboma Diriba.

**Methodology:** Tadesse Sime, Lemessa Oljira, Aboma Diriba, Wubishet Gezimu.

**Project administration:** Tadesse Sime, Lemessa Oljira.

**Resources:** Tadesse Sime, Lemessa Oljira, Aboma Diriba, Gamachis Firdisa, Wubishet Gezimu.

**Software:** Tadesse Sime, Lemessa Oljira, Gamachis Firdisa, Wubishet Gezimu.

**Supervision:** Tadesse Sime, Lemessa Oljira, Gamachis Firdisa.

**Validation:** Tadesse Sime, Lemessa Oljira, Aboma Diriba.

**Visualization:** Tadesse Sime, Lemessa Oljira, Gamachis Firdisa, Wubishet Gezimu.

**Writing – original draft:** Tadesse Sime, Gamachis Firdisa, Wubishet Gezimu.

**Writing – review & editing:** Tadesse Sime, Aboma Diriba, Gamachis Firdisa, Wubishet Gezimu.

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
