## [Decision Letter · Decision Letter 0]

6 Sep 2022

PONE-D-22-21893Effect of active tuberculosis on the survival of HIV-infected adult patients who initiated antiretroviral therapy at public hospitals of Eastern Ethiopia: A retrospective cohort studyPLOS ONE

Dear Dr. Gezimu,

Thank you for submitting your manuscript to PLOS ONE. After careful consideration, we feel that it has merit but does not fully meet PLOS ONE’s publication criteria as it currently stands. Therefore, we invite you to submit a revised version of the manuscript that addresses the points raised during the review process.

We look forward to receiving your revised manuscript.

Kind regards,

Ari Samaranayaka, PhD

Academic Editor

PLOS ONE

Journal Requirements:

2. We note you have included a table to which you do not refer in the text of your manuscript. Please ensure that you refer to Table 2 in your text; if accepted, production will need this reference to link the reader to the Table.

Additional Editor Comments:

Major comments:

1. Although authors say all data are within supporting files, supporting documents include the questionnaire and strobe list, not the data.

2. line 89-91. please rephrase to improve clarify. I assumed people who started ART treatment during these times points were included in the study, and classified into infected and not-infected groups.

3. line 96. How the sample size was calculated was explained, but the resultant sample size was not given, why? Appears N=566 recruitment was not based on above calculation.

4. Please explain patients identifying procedure more clearly. If people were randomly selected from health records, then their TB-infection status was determined, how did you ensure equal sized groups (N=283 in both TB-positive and TB-negative groups)?

5. line 114. “All the profiles of HIV-infected patients between January 1, 2014, and June 30, 2018, were reviewed”. why you review all patients, instead of reviewing only those selected, if patients were selected using simple random sampling?

6. Cox regression was used to analyse the duration from ART initiation to the death. What is the definition of death? Line 131 says “all-cause mortality”, but line 313 says disease-specific mortality.

7. line 168. “Kaplan-Meier test”. Is it correct to describe this estimation procedure as a test?

8. line 170. “The log-rank test was used to compare the median time to death between patients with TB and those without TB”. Can logrank test compare median times to death?

9. By looking at figure1 I assume proportional assumptions is ok, but for completeness please include in statistical analysis section how you assessed it.

10. tables 1 and 2. I think you have presented ttest statistics as chisquare statistics. Also, explain why test statistic not given for some; eg, fishers exact was used?

11. “median time to death was 2.5 months and 12.2 months in the TB co-infected cohort and non-TB co-infected cohort, respectively”. How many deaths were used in each of these groups to arrive at these estimates?

12. table3. Why N was reduced when there were no missing values in of the variables?

Minor comments

1. undefined abbreviations. eg; ART, PLWH, PMTCT, OPI, PY, spell out all abbreviations at first use.

2. line 54. what is meant by 5.9 patients, is it 5.9% ?.

3. line 57. why two percentages active TB patients (14.2% and 11.5%) for Harari?

4. line 68-72. Cannot understand this very long sentence with undefined abbreviations.

5. line 117. “a baseline” or “the baseline”?

6. Line 130. operational.

7. table2 title. repetition.

8. reference 19. what is meant by “Vol 618” when this is a text book?

Reviewers' comments:

Reviewer's Responses to Questions

**Comments to the Author**

1. Is the manuscript technically sound, and do the data support the conclusions?

Reviewer #1: Yes

Reviewer #2: Yes

Reviewer #3: Partly

Reviewer #4: Yes

2. Has the statistical analysis been performed appropriately and rigorously? 

Reviewer #1: Yes

Reviewer #2: Yes

Reviewer #3: Yes

Reviewer #4: Yes

3. Have the authors made all data underlying the findings in their manuscript fully available?

Reviewer #1: Yes

Reviewer #2: Yes

Reviewer #3: Yes

Reviewer #4: Yes

4. Is the manuscript presented in an intelligible fashion and written in standard English?

Reviewer #1: Yes

Reviewer #2: Yes

Reviewer #3: Yes

Reviewer #4: No

5. Review Comments to the Author

Reviewer #1: 1. Is the manuscript technically sound, and do the data support the conclusions?

Yes, the manuscript is technically sound. The conclusions were drawn appropriately based ob the data presented. The retrospective study had experimental and control groups. The sample size was adequate from the power analysis of the study.

What is our baseline definition for this study? Is it tuberculosis co-infection at antiretroviral therapy initiation? If yes, clarify the WHO clinical staging for the experimental group with respect to TB co-infection. Lines 205 and 206 talked about 71.4% of TB co-infected patient group as being in WHO clinical stage III at ART initiation. By definition presence of TB infection automatically places a patient in at least clinical stage III. 4.6% of patients were reported to be in clinical stages I and I. Why do we have stages I and II with TB-confection for patients at baseline for the experimental group.

2. Has the statistical analysis been performed appropriately and rigorously?

Yes, the statistical analyses were appropriate for the study

3. Have the authors made all data underlying the findings in their manuscript fully available?

Yes, all data underlying the findings in the manuscript were made available by the authors

4. Is the manuscript presented in an intelligible fashion and written in standard English?

Some typos and grammatical errors were found.

Here are a few examples I found:

Line 130 - typo on "Operational"

Line 141 - correct word is "GeneXpert" not Gen Xpert

Line 150 - correct word is "Co-trimoxazole"

205 two cohorts. The majority of TB co-infected group patients were in WHO clinical stage III at. "Patients should come before group" corrected should be "The majority of TB co-infected patients group were in WHO clinical stage III at"

Line 305 "Africa, which faster prognosis of disease and increase opportunistic infection" is grammatically wrong

Reviewer #2: This study by Wubishet Gezimu, et al., was an institution-based retrospective cohort conducted in public hospitals in Eastern Ethiopia. The participants were randomly selected from January 1, 2014 to June 30, 2018.

The purpose of the study was to determine the effect of active tuberculosis on the survival of HIV-infected adult patients who initiated antiretroviral therapy in public hospitals in Eastern Ethiopia.

Dependent (outcome) variables: Time from ART initiation to death

Independent (predictors) variables:

• sociodemographic features: age, sex, place of residence, marital status, educational status, occupation, and clinical.

• treatment-related characteristics: WHO clinical stage, CD4 count level, hemoglobin level, adherence to ART, co-trimoxazole prophylaxis status, BMI, and functional status

• opportunistic infections other than TB were included.

A total of 566 participants were randomly selected from January 1, 2014, to June 30, 2018. The collected data were entered into EpiData version 3.1 and then exported to Stata version 14

software for analysis. A Cox proportional hazard model was used to determine the

effect of active tuberculosis on the survival of HIV-infected adult patients who initiated

antiretroviral therapy, and a p-value less than 0.05 and a 95% confidence level was

used to declare statistical significance. Ethical clearance was obtained from the Institutional Health Research and Ethical Review Committee (IHRERC) of Haramaya University College of health and medical sciences.

The results revealed that of the 566 patients included in the study, 76 died. The mortality rate was 11.04 per 100 person-years in tuberculosis co-infected patient while 2.52 per 100 person years in non-tuberculosis co-infected patients. The patients with tuberculosis coinfection

had a 2.19 times higher hazard of death (AHR: 2.19; 95% CI: 1.17, 4.12) compared to those without tuberculosis. Advanced clinical stage, low CD4+ cell count, and previous episodes of an opportunistic infection other than tuberculosis were discovered to be independent predictors of mortality.

The authors made all data underlying the findings fully available. The data was tested for representativeness, analyzed using descriptive and inferential statistics which were rigorous and appropriate.

Discussions of the results were robust, citing similar studies conducted both within and outside Ethiopia.

Conclusions are in line with the findings

Writing quality and clarity requires editing especially the conclusion: e.g.

In this study showed that HIV patients who TB co-infected at ART initiation were experienced substantial mortality rates…………. should read: This study showed that HIV patients who are co-infected with TB at ART initiation experienced substantial mortality rates etc.

Other observations:

1. Limitations of the study: The authors did well to mention the limitations of the study

2. Inclusion/exclusion criteria clearly explained as a separate sub topic

Reviewer #3: The author in their manuscript 'Effect of active tuberculosis on the survival of HIV-infected adult patients who initiated

antiretroviral therapy at public hospitals of Eastern Ethiopia: A retrospective cohort study' elaborated the need of strengthening the tuberculosis screening among HIV positive patients by showing higher death rate of among HIV-TB coinfection that could be prevented by rigorous early screening.

However, there are some concern in the current manuscript.

1. The sample size is small, though the conclusion/outcome may not change with the increased sample size but the rigourness of data analysis will certainly increase.

2. The findings are not novel. Many studies already conducted in similar set up including in Ethiopia with similar outcome. In the current manuscript no new or novel information is added.

3. The discussion section need to be re written as there are repetition and redundancy in writing.

Reviewer #4: Abstract: Consider starting abstract with "In resource limited setting such as Ethiopia,.....". TB is a major cause of morbidity and mortality among PLHIV but the current sentence structure states that TB is a major cause of mortality in resource limited setting; the context for the analysis is among PLHIV. Likewise, the sentence structure needs to be adjusted throughout the manuscript.

Introduction: Spell out TB when first used earlier in the introduction rather than spelling it out in line 48. Grammatical error in line 55-57 makes sentence difficult to understand.

Line 70:spell out acronyms when first used in the paper eg. PLWH, HAART

Line 91: punctuation error. Review appropriateness of punctuations throughout the manuscript

6. PLOS authors have the option to publish the peer review history of their article (what does this mean?). If published, this will include your full peer review and any attached files.

Reviewer #1: No

Reviewer #2: **Yes: **Haruna Ismaila ADAMU, MBBS; MPH; PhD

Reviewer #3: No

Reviewer #4: No

---

## [Author Response · Author response to Decision Letter 0]

21 Sep 2022

Response to Reviewers

Manuscript ID: PONE-D-22-21893

Title: Effect of active tuberculosis on the survival of HIV-infected adult patients who initiated antiretroviral therapy at public hospitals of Eastern Ethiopia: A retrospective cohort study

Dear Editor and Reviewers, We are so pleased with your prompt feedback on our article. Thanks for your valuable and intellectual comments and suggestions. We have corrected and responded to all the comments and suggestions raised by the editor and each reviewer, point by point, hereunder.

Journal Requirements:

Author responses: Dear thanks so much for your suggestion. We have revised all our submissions as per the PLOS ONE’s styles. 

2. We note you have included a table to which you do not refer in the text of your manuscript. Please ensure that you refer to Table 2 in your text; if accepted, production will need this reference to link the reader to the Table.

Author responses: Thank you very much for your important suggestion. We are sorry for the editorial error. We have just corrected it (mentioned table 2 in the text). 

Additional Editor Comments:

Major comments:

1. Although authors say all data are within supporting files, supporting documents include the questionnaire and strobe list, not the data.

Author responses: Dear Editor, Thank you so much for your suggestion. We have included the data used for this study within the revision file in the zipped supportive information file.

2. line 89-91. Please rephrase to improve clarify. I assumed people who started ART treatment during these times points were included in the study, and classified into infected and not-infected groups.

Author responses: Dear Editor, Thanks for your intellectual and constructive comments. The concept is exactly what you mentioned above. We have rephrased it as follows: PLHIV who started ART from January 1, 2014, to June 30, 2018, were included in the study and classified into tuberculosis co-infected and not-infected groups. Patients with TB co-infected were included in the exposed group, whereas those not co-infected with TB were included in the non-exposed group.

3. line 96. How the sample size was calculated was explained, but the resultant sample size was not given, why? Appears N=566 recruitment was not based on above calculation. 

Author responses: Dear Editor Thanks a lot for your constructive comments. Considering the mentioned assumption, the calculated final sample size was 566. Finally, a minimum sample sizes of 566 was used for this study, which means 283 in the exposed cohort and 283 in the non-exposed cohort were assigned randomly.

4. Please explain patients identifying procedure more clearly. If people were randomly selected from health records, then their TB-infection status was determined, how did you ensure equal sized groups (N=283 in both TB-positive and TB-negative groups)?

Author responses: Dear Editor, thanks for your important comment. From January 1, 2014, to June 30, 2018, a total of 1785 patients initiated ART in public hospitals in Harar and Dire Dawa towns. Of these, 115 patients (10 who started ART for PMTCT, 25 with incomplete data, and 80 who transferred in) were excluded because they were ineligible. The ART registration and follow-up forms were used to determine the patient's TB status at ART initiation. Patients were categorized as exposed or non-exposed based on their TB status. Finally, a proportional allocation followed by a simple random sampling was applied to select TB co-infected and not-co-infected patients from each hospital. We have also briefly presented this section in Figure 1.

5. Line 114. “All the profiles of HIV-infected patients between January 1, 2014, and June 30, 2018, were reviewed”. why you review all patients, instead of reviewing only those selected, if patients were selected using simple random sampling?

Author responses: Dear Editor, Thank you for your insightful comments. Sorry for the editorial error. We have corrected it. As you exactly mentioned, the profiles of all selected HIV-infected patients between January 1, 2014, and June 30, 2018, were reviewed.

6. Cox regression was used to analyse the duration from ART initiation to the death. What is the definition of death? Line 131 says “all-cause mortality”, but line 313 says disease-specific mortality.

Author responses: Dear Editor, Thank you so much for your scholarly suggestion. For this study, we considered all-cause mortality while the patient is on ART. The death of any patient while on ART is reported as an HIV death since it is difficult to get cause-specific data. To consider competing for risk analysis, we tried to identify the cause of death. Fortunately, on our ascertainment from the patient's death certificate and through telephone by ART adherence supporters, all deaths were due to illness and not attributable to accidents or other unnatural causes. 

7. line 168. “Kaplan-Meier test”. Is it correct to describe this estimation procedure as a test?

Author responses: Dear Editor, I really thank you very much for such an important comment. It was an editorial error. We have just corrected it as a Kaplan-Meier survival estimate.

8. line 170. “The log-rank test was used to compare the median time to death between patients with TB and those without TB”. Can logrank test compare median times to death?

Author responses: Dear Editor, thank you very much for such a productive comment. The log-rank test was used to test the equality of survival functions between patients with TB and those without TB.

9. By looking at figure1 I assume proportional assumptions is ok, but for completeness please include in statistical analysis section how you assessed it.

Author responses: Dear Editor Thanks a lot for your comment. Cox proportional hazard model fitness was checked using graphically and schoenfield residuals test. The resulting curves were parallel, and schoenfield residuals test was statistically not significant. Thus, none of the predictors violated the proportional hazard assumption. We have just explained the mentioned section in the statistical analysis section.

10. tables 1 and 2. I think you have presented ttest statistics as chisquare statistics. Also, explain why test statistic not given for some; eg, fishers exact was used?

Author responses: Dear Editor Thanks so much for your insightful comment. The Chi-square and T-test statistics were used to compare categorical and continuous variables between the two cohorts, respectively. We have used the fishers exact in case of the Chi-squared test statistics is invalid (when about 20% of the expected cell has less than value of 5), 

11. “median time to death was 2.5 months and 12.2 months in the TB co-infected cohort and non-TB co-infected cohort, respectively”. How many deaths were used in each of these groups to arrive at these estimates? 

Author responses: Dear Thanks a lot. Of the 76 total deaths, 60 patients were from TB infected and 16 were died from not TB infected group. 

12. table3. Why N was reduced when there were no missing values in of the variables?

Author responses: Dear Editor Thank you very much for such an important suggestion. As we have also mentioned in the descriptive statistics part (in table 2), 14 patients data were missed on the baseline CD4+ cell count. Consequently, all variables included in the Cox regression analysis have been reduced to N =552. 

Minor comments

1. undefined abbreviations. eg; ART, PLWH, PMTCT, OPI, PY, spell out all abbreviations at first use.

Author responses: Dear Editor Thanks for your comments. We have just spelled out all the mentioned abbreviations and others at their first use (ART=antiretroviral therapy, PLWH=people living with HIV, PMTCT=Prevention of Mother to Child Transmission, OPI= opportunistic infection, PY: Person year). 

2. line 54. what is meant by 5.9 patients, is it 5.9% ?.

Author responses: Dear Editor Thanks a lot. As you mentioned, it is 5.9%. We have corrected it.

3. line 57. why two percentages active TB patients (14.2% and 11.5%) for Harari?

Author responses: Dear Editor Thanks for your most valuable comment. It was editorial error. We have just corrected the section. According to the Ethiopia’s 2014 TB-HIV surveillance report, 9.1% of HIV-infected clients newly enrolled in HIV care had active TB. The percentages of active TB patients in Harar Regional State and Dire Dawa City Administration were 14.2% and 11.5%, respectively. 

4. line 68-72. Cannot understand this very long sentence with undefined abbreviations.

Author responses: Dear Editor Thank you so much for your important suggestion. We have corrected the mentioned sentence. The effect of active TB co-infection on the survival of HIV patients in the ART era remains inconclusive, and mortality among TB-HIV co-infected patients were strongly associated with the absence of ART. 

5. line 117. “a baseline” or “the baseline”?

Author responses: Dear Editor Thanks a lot for you’re an important suggestion. We have just corrected the mentioned section. It was to mean the baseline. 

6. Line 130. operational.

Author responses: Dear Editor Thanks so much. We have corrected the mentioned typo.

7. table2 title. repetition.

Author responses: Dear Editor Thanks for your suggestion. We have just corrected the mentioned section. 

8. reference 19. what is meant by “Vol 618” when this is a text book?

Author responses: Dear Editor Thank you so much for the suggestion. We are sorry for the editorial error. We have corrected mentioned reference.

Reviewer #1

What is our baseline definition for this study? Is it tuberculosis co-infection at antiretroviral therapy initiation? If yes, clarify the WHO clinical staging for the experimental group with respect to TB co-infection. Lines 205 and 206 talked about 71.4% of TB co-infected patient group as being in WHO clinical stage III at ART initiation. By definition presence of TB infection automatically places a patient in at least clinical stage III. 4.6% of patients were reported to be in clinical stages I and I. Why do we have stages I and II with TB-confection for patients at baseline for the experimental group.

Author response: Dear Reviewer, Thank you very much for your knowledgeable comment. Indeed, as you pointed out, the presence of TB co-infection automatically places a patient in clinical stage III or IV. However, in our study, TB co-infection was considered within the first three months of ART initiation. During the document review, we considered the patients registered as clinical stage I or II at baseline, from which some (4.6%) of patients developed TB within the first three months of ART initiation. 

Line 130 - typo on "Operational"

Author responses: Dear Reviewer Thanks so much for your important comment. We have corrected the mentioned typo.

Line 141 - correct word is "GeneXpert" not Gen Xpert

Author responses: Dear Reviewer Thanks a lot. We have corrected the mentioned section. 

Line 150 - correct word is "Co-trimoxazole"

Author responses: Dear reviewer, thank you very much for your comment. We have corrected the mentioned term. 

205 two cohorts. The majority of TB co-infected group patients were in WHO clinical stage III at. "Patients should come before group" corrected should be "The majority of TB co-infected patients group were in WHO clinical stage III at"

Author responses: Dear Reviewer Thank you so much. We have corrected the mentioned sentence. 

Line 305 "Africa, which faster prognosis of disease and increase opportunistic infection" is grammatically wrong

Author responses: Thanks so much. We have just corrected the grammar error in the mentioned sentence. 

Reviewer #2: 

This study by Wubishet Gezimu, et al., was an institution-based retrospective cohort conducted in public hospitals in Eastern Ethiopia. The participants were randomly selected from January 1, 2014 to June 30, 2018.

The purpose of the study was to determine the effect of active tuberculosis on the survival of HIV-infected adult patients who initiated antiretroviral therapy in public hospitals in Eastern Ethiopia.

Dependent (outcome) variables: Time from ART initiation to death

Independent (predictors) variables:

• sociodemographic features: age, sex, place of residence, marital status, educational status, occupation, and clinical.

• treatment-related characteristics: WHO clinical stage, CD4 count level, hemoglobin level, adherence to ART, co-trimoxazole prophylaxis status, BMI, and functional status

• opportunistic infections other than TB were included.

A total of 566 participants were randomly selected from January 1, 2014, to June 30, 2018. The collected data were entered into EpiData version 3.1 and then exported to Stata version 14 software for analysis. A Cox proportional hazard model was used to determine the effect of active tuberculosis on the survival of HIV-infected adult patients who initiated antiretroviral therapy, and a p-value less than 0.05 and a 95% confidence level was used to declare statistical significance. Ethical clearance was obtained from the Institutional Health Research and Ethical Review Committee (IHRERC) of Haramaya University College of health and medical sciences.

The results revealed that of the 566 patients included in the study, 76 died. The mortality rate was 11.04 per 100 person-years in tuberculosis co-infected patient while 2.52 per 100 person years in non-tuberculosis co-infected patients. The patients with tuberculosis coinfection had a 2.19 times higher hazard of death (AHR: 2.19; 95% CI: 1.17, 4.12) compared to those without tuberculosis. Advanced clinical stage, low CD4+ cell count, and previous episodes of an opportunistic infection other than tuberculosis were discovered to be independent predictors of mortality.

The authors made all data underlying the findings fully available. The data was tested for representativeness, analyzed using descriptive and inferential statistics which were rigorous and appropriate.

Author response: Dear Reviewer Thanks for your intellectual and constructive comments.

Discussions of the results were robust, citing similar studies conducted both within and outside Ethiopia.

Author response: Dear Reviewer Thanks for your intellectual and constructive comments.

Conclusions are in line with the findings

Author response: Dear Reviewer Thanks for your intellectual and constructive comments.

Writing quality and clarity requires editing especially the conclusion: e.g.

In this study showed that HIV patients who TB co-infected at ART initiation were experienced substantial mortality rates…………. should read: This study showed that HIV patients who are co-infected with TB at ART initiation experienced substantial mortality rates etc.

Authors response: Dear Reviewer Thanks for your important suggestion. We have corrected the mentioned section as ‘’This study showed that HIV patients who were co-infected with TB at ART initiation experienced substantial mortality rates’’

Other observations:

1. Limitations of the study: The authors did well to mention the limitations of the study

Author response: Dear Reviewer, Thank you so much for your constructive comments. As you mentioned, we have clearly stated the limitations of our study.

2. Inclusion/exclusion criteria clearly explained as a separate sub topic

Author response: Thanks once again. We have clearly explained the inclusion and exclusion criteria as a separate subtopic.

Reviewer #3: 

The author in their manuscript 'Effect of active tuberculosis on the survival of HIV-infected adult patients who initiated antiretroviral therapy at public hospitals of Eastern Ethiopia: A retrospective cohort study' elaborated the need of strengthening the tuberculosis screening among HIV positive patients by showing higher death rate of among HIV-TB coinfection that could be prevented by rigorous early screening.

However, there are some concerns in the current manuscript.

1. The sample size is small, though the conclusion/outcome may not change with the increased sample size but the rigourness of data analysis will certainly increase.

Authors’ response: Dear reviewer thank you for your intellectual comment. Indeed the sample size was not large enough. However, it was calculated by using the probability of event by considering the interval in the enrolment period. And also appropriate scientific assumptions were utilized in the sample size calculation. Beside these, the sample was adequately distributed to all public hospitals included in the study. Additionally, we have conducted rigorous data analysis. Hence, we have no any doubt regarding the study’s conclusion. 

2. The findings are not novel. Many studies already conducted in similar set up including in Ethiopia with similar outcome. In the current manuscript no new or novel information is added.

Authors’ response: Dear reviewer thank you for your scholarly comment. Actually, many studies were conducted in Ethiopia. A very important issue that makes the current study unique is that it was conducted at the era when an antiretroviral therapy is initiated at the time of HIV diagnosis (regardless of the patients’ CD4+ cell count). In contrary to this, many of the previous studies from Ethiopia were conducted at the era when ART initiation was based on the patients’ CD4+ cell count. Therefore, all participants were ART cohort (the effect of ART was controlled for both groups) in the current study. Additionally, as far as our knowledge, no study has been conducted in the current study area considering two explicit groups of exposed and unexposed. In the current study, both exposed and unexposed groups have an equal size to compare exposed with unexposed group. 

3. The discussion section need to be re written as there are repetition and redundancy in writing.

Authors’ response: Dear Reviewer, Thank you so much for your helpful suggestions. We have just rewritten the discussion section. Thanks for your time with our manuscript.

Reviewer #4: 

Abstract: Consider starting abstract with "In resource limited setting such as Ethiopia,.....". TB is a major cause of morbidity and mortality among PLHIV but the current sentence structure states that TB is a major cause of mortality in resource limited setting; the context for the analysis is among PLHIV. Likewise, the sentence structure needs to be adjusted throughout the manuscript.

Authors response: Dear Reviewer, Thanks for such an intelligent suggestion. We have corrected the structure of the mentioned section in the abstract and all over the document. 

Introduction: Spell out TB when first used earlier in the introduction rather than spelling it out in 

Author responses: Dear Reviewer, Thank you so much for your valuable suggestion. We have spelled out TB in first time use.

line 48. Grammatical error in line 55-57 makes sentence difficult to understand.

Authors Response: Dear reviewer thanks once again. We have corrected the mentioned section and currently it is understandable.

Line 70: spell out acronyms when first used in the paper eg. PLWH, HAART

Authors Response: Dear Reviewer Thanks a lot. We have spelled out the abbreviations. NOTE: We have replaced ‘’HAART’’ with ‘’ART’’ just to be consistent.

Line 91: punctuation error. Review appropriateness of punctuations throughout the manuscript

Authors Response: Dear Reviewer, Thanks for your suggestion. We have corrected the punctuation errors in the mentioned section and all throughout the manuscript.

---

## [Decision Letter · Decision Letter 1]

11 Oct 2022

PONE-D-22-21893R1Effect of active tuberculosis on the survival of HIV-infected adult patients who initiated antiretroviral therapy at public hospitals of Eastern Ethiopia: A retrospective cohort studyPLOS ONE

Dear Dr. Gezimu,

Thank you for submitting your manuscript to PLOS ONE. After careful consideration, we feel that it has merit but does not fully meet PLOS ONE’s publication criteria as it currently stands. Therefore, we invite you to submit a revised version of the manuscript that addresses the points raised during the review process.

We look forward to receiving your revised manuscript.

Kind regards,

Ari Samaranayaka, PhD

Academic Editor

PLOS ONE

Journal Requirements:

Additional Editor Comments:

Newly added Figure 1 explain the recruitment procedure, it is clear. Required sample size of N=566 was proportionally divided between 4 hospitals. Then number required from in each hospital was divided equally between exposed and unexposed groups. Then those 8 numbers were recruited randomly. Therefore it is incorrect to describe the sampling process as simple random sampling, or as random selection from each hospital. They have used a type of stratified sampling (random sampling stratified by hospital and exposure status).

Reviewers' comments:

Reviewer's Responses to Questions

**Comments to the Author**

1. If the authors have adequately addressed your comments raised in a previous round of review and you feel that this manuscript is now acceptable for publication, you may indicate that here to bypass the “Comments to the Author” section, enter your conflict of interest statement in the “Confidential to Editor” section, and submit your "Accept" recommendation.

Reviewer #2: All comments have been addressed

Reviewer #3: All comments have been addressed

Reviewer #4: All comments have been addressed

2. Is the manuscript technically sound, and do the data support the conclusions?

Reviewer #2: Yes

Reviewer #3: Yes

Reviewer #4: Yes

3. Has the statistical analysis been performed appropriately and rigorously? 

Reviewer #2: Yes

Reviewer #3: Yes

Reviewer #4: Yes

4. Have the authors made all data underlying the findings in their manuscript fully available?

Reviewer #2: Yes

Reviewer #3: Yes

Reviewer #4: Yes

5. Is the manuscript presented in an intelligible fashion and written in standard English?

Reviewer #2: Yes

Reviewer #3: Yes

Reviewer #4: No

6. Review Comments to the Author

Reviewer #2: The authors have taken care of all the observations and suggestions I made during my first review. The paper should be ready for publication, subject to the satisfaction of the other reviewers and the editor.

Reviewer #3: The authors has adequately responded to the queries raised by the reviewer.

However, at the start of the discussion section some modification may be needed as commented below.

In the current study, the overall mortality rate was 6.55 per 100 PYO in the entire follow-up

period. This finding is “comparable with a previous study conducted in Ethiopia”[13]; [11].

However, it is “higher compared to the studies conducted in Ethiopia” and other countries of Africa.

The part of the sentences … “comparable with a previous study conducted in Ethiopia”.. and … “higher compared to the studies conducted in Ethiopia”…. are seems to be in disagreement and may need to be fixed.

Reviewer #4: Manuscript still has some grammatical errors that need to be corrected.

Lines 249 to 251 offers conflicting information for instance

7. PLOS authors have the option to publish the peer review history of their article (what does this mean?). If published, this will include your full peer review and any attached files.

Reviewer #2: **Yes: **Haruna Ismaila ADAMU MBBS; MPH; PhD

Reviewer #3: No

Reviewer #4: No

---

## [Author Response · Author response to Decision Letter 1]

14 Oct 2022

Response to Reviewers

Manuscript ID: PONE-D-22-21893R1

Title: Effect of active tuberculosis on the survival of HIV-infected adult patients who initiated antiretroviral therapy at public hospitals of Eastern Ethiopia: A retrospective cohort study

Respected Reviewers and Editors, We are really grateful for your quick response to our article. We appreciate your considerate and insightful comments and recommendations. Each comment and recommendation made by the editor and each reviewer has been addressed in the following sections with corrections and responses.

Journal Requirements:

Please review you reference list to ensure that it is complete and correct. If you have cited papers that have been retracted, please include the rationale for doing so in the manuscript text, or remove these references and replace them with relevant current references. Any changes to the reference list should be mentioned in the rebuttal letter that accompanies your revised manuscript. If you need to cite a retracted article, indicate the article’s status in the References list and also include a citation and full reference for the retraction notice. 

Author response: Thank you so much for your important suggestion. We have checked the reference list for its completeness and correctness. We have not cited any retracted articles. We have cited reference number five (previously missed) in our discussion section of paragraph 4 (Page 17, line no. 276). Additionally, we have corrected reference 12. Thanks once again.

Additional Editor Comments:

Newly added Figure 1 explain the recruitment procedure, it is clear. Required sample size of N=566 was proportionally divided between 4 hospitals. Then number required from in each hospital was divided equally between exposed and unexposed groups. Then those 8 numbers were recruited randomly. Therefore it is incorrect to describe the sampling process as simple random sampling, or as random selection from each hospital. They have used a type of stratified sampling (random sampling stratified by hospital and exposure status). 

Author response: Dear Editor, We are so grateful for your intellectual suggestions. Thanks a lot. What you mentioned is all about our sampling technique. We have corrected the previous description of the sampling technique as per your suggestion. As you stated, we used a random sampling stratified by hospital, and exposure status was used to select the study participants. First, the required sample was proportionally allocated to the four hospitals. Then the population in each hospital was stratified based on their exposure status. Finally, each study participant was selected randomly from each stratum. Thanks once again.

Reviewer #2

The authors have taken care of all observations and suggestions I made during my first review. The paper should be ready for publication, subject to the satisfaction of the other reviewers and the editor. 

Author response: Dear Reviewer, Thanks for your time with our work. 

Reviewer #3

The authors has adequately responded to the queries raised by the reviewer.

However, at the start of the discussion section some modification may be needed as commented below.

In the current study, the overall rate was 6.55 per 100 PYO in the entire follow-up period This finding is “comparable with a previous study conducted in Ethiopia”([13]; [11].

However, it is “higher compared to the studies conducted in Ethiopia” and other countries of African.

The part of these sentences…”comparable with a previous study conducted in Ethiopia”.. and …“higher compared to the studies conducted in Ethiopia” … are seems to be in disagreement and may need to be fixed. 

Author response: Dear Reviewer, Thank you so much for your rational comment. We have corrected the mentioned section by specifying the particular study settings of those previous studies conducted in Ethiopia. It is corrected as “This finding is comparable with the previous studies conducted in the seven University Teaching Hospitals and Somali Regional State of Ethiopia[13]; [11].” “However, it is higher compared to the studies conducted in Jinka Hospital, South Omo zone of Ethiopia and other countries of Africa [14]; [20] : [21].”

Reviewer #4

Manuscript still has some grammatical errors that need to be corrected. 

Author response: Dear Reviewer, Thanks for your suggestion. We have corrected all grammar errors throughout the manuscript. 

Lines 241 to 251 offers conflicting information for instance. 

Author response: Dear Reviewer, Thanks so much for you intellectual comments. We have corrected the mentioned section by specifying the particular study areas of those previous studies conducted in Ethiopia. It is corrected as “This finding is comparable with the previous studies conducted in the seven University Teaching Hospitals and Somali Regional State of Ethiopia[13]; [11].” “However, it is higher compared to the studies conducted in Jinka Hospital, South Omo zone of Ethiopia and other countries of Africa [14]; [20] : [21].”

---

## [Editor Report · Decision Letter 2]

18 Oct 2022

Effect of active tuberculosis on the survival of HIV-infected adult patients who initiated antiretroviral therapy at public hospitals of Eastern Ethiopia: A retrospective cohort study

PONE-D-22-21893R2

Dear Dr. Gezimu,

We’re pleased to inform you that your manuscript has been judged scientifically suitable for publication and will be formally accepted for publication once it meets all outstanding technical requirements.

Kind regards,

Ari Samaranayaka, PhD

Academic Editor

PLOS ONE
---

## [Editor Report · Acceptance letter]

21 Oct 2022

PONE-D-22-21893R2 

Effect of active tuberculosis on the survival of HIV-infected adult patients who initiated antiretroviral therapy at public hospitals of Eastern Ethiopia: A retrospective cohort study 

Dear Dr. Gezimu:

I'm pleased to inform you that your manuscript has been deemed suitable for publication in PLOS ONE. Congratulations! Your manuscript is now with our production department. 

Kind regards, 

on behalf of

Dr. Ari Samaranayaka 

Academic Editor

PLOS ONE